# Silibinin Suppresses the Hyperlipidemic Effects of the ALK-Tyrosine Kinase Inhibitor Lorlatinib in Hepatic Cells

**DOI:** 10.3390/ijms23179986

**Published:** 2022-09-01

**Authors:** Sara Verdura, José Antonio Encinar, Salvador Fernández-Arroyo, Jorge Joven, Elisabet Cuyàs, Joaquim Bosch-Barrera, Javier A. Menendez

**Affiliations:** 1Metabolism and Cancer Group, Program Against Cancer Therapeutic Resistance (ProCURE), Catalan Institute of Oncology, 17007 Girona, Spain; 2Girona Biomedical Research Institute (IDIBGI), Salt, 17190 Girona, Spain; 3Institute of Research, Development and Innovation in Biotechnology of Elche (IDiBE) and Molecular and Cell Biology Institute (IBMC), Miguel Hernández University (UMH), 03207 Elche, Spain; 4Department of Medicine and Surgery, Universitat Rovira i Virgili, 43204 Reus, Spain; 5Unitat de Recerca Biomèdica (URB-CRB), Hospital Universitari de Sant Joan, Institut d’Investigació Sanitaria Pere Virgili, Universitat Rovira i Virgili, 43204 Reus, Spain; 6Medical Oncology, Catalan Institute of Oncology, 17007 Girona, Spain; 7Department of Medical Sciences, Medical School, University of Girona, 17071 Girona, Spain

**Keywords:** lorlatinib, hypertriglyceridemia, hypercholesteremia, silibinin, lipidomics, CYP3A4, statins

## Abstract

The third-generation anaplastic lymphoma tyrosine kinase inhibitor (ALK-TKI) lorlatinib has a unique side effect profile that includes hypercholesteremia and hypertriglyceridemia in >80% of lung cancer patients. Here, we tested the hypothesis that lorlatinib might directly promote the accumulation of cholesterol and/or triglycerides in human hepatic cells. We investigated the capacity of the hepatoprotectant silibinin to modify the lipid-modifying activity of lorlatinib. To predict clinically relevant drug–drug interactions if silibinin were used to clinically manage lorlatinib-induced hyperlipidemic effects in hepatic cells, we also explored the capacity of silibinin to interact with and block CYP3A4 activity using in silico computational descriptions and in vitro biochemical assays. A semi-targeted ultrahigh pressure liquid chromatography accurate mass quadrupole time-of-flight mass spectrometry with electrospray ionization (UHPLC-ESI-QTOF-MS/MS)-based lipidomic approach revealed that short-term treatment of hepatic cells with lorlatinib promotes the accumulation of numerous molecular species of cholesteryl esters and triglycerides. Silibinin treatment significantly protected the steady-state lipidome of hepatocytes against the hyperlipidemic actions of lorlatinib. Lipid staining confirmed the ability of lorlatinib to promote neutral lipid overload in hepatocytes upon long-term exposure, which was prevented by co-treatment with silibinin. Computational analyses and cell-free biochemical assays predicted a weak to moderate inhibitory activity of clinically relevant concentrations of silibinin against CYP3A4 when compared with recommended (rosuvastatin) and non-recommended (simvastatin) statins for lorlatinib-associated dyslipidemia. The elevated plasma cholesterol and triglyceride levels in lorlatinib-treated lung cancer patients might involve primary alterations in the hepatic accumulation of lipid intermediates. Silibinin could be clinically explored to reduce the undesirable hyperlipidemic activity of lorlatinib in lung cancer patients.

## 1. Introduction

The small macrocyclic compound lorlatinib (product name PF-06463922) is a third-generation anaplastic lymphoma kinase (ALK) tyrosine kinase inhibitor (TKI) originally developed to inhibit ALK mutant forms causing resistance to first- and second-generation ALK-TKIs (e.g., crizotinib, ceritinib, alectinib or brigatinib) in patients with advanced non-small-cell lung cancer (NSCLC) [1,2,3,4,5,6]. Consistent with its broad ALK mutational coverage and optimized central nervous system penetration through the blood–brain barrier, lorlatinib has shown substantial systemic and intracranial activity both in treatment-naïve patients and in patients with relapse after first- and second-generation ALK TKIs [7,8,9,10,11,12]. It has been approved for the treatment of patients with ALK-positive metastatic NSCLC whose disease has progressed after alectinib or ceritinib as the first ALK-TKI, or after crizotinib and at least one other ALK-TKI [13]. 

Lorlatinib has a unique treatment-related adverse event profile compared with other ALK-TKIs that is characterized by the frequent incidence of hypercholesteremia and hypertriglyceridemia, which have been found to occur in 61% and 82% of lorlatinib-treated patients included in a pooled safety analysis (*n* = 295) [14,15]. Results from the global phase 2 study B7461001 (NCT01970865) revealed that grade 3 or 4 elevations in total cholesterol and triglycerides both occurred at a frequency of 16% [2]. Hyperlipidemia—comprising the cluster terms hypercholesterolemia and hypertriglyceridemia—is the most commonly reported adverse reaction to lorlatinib, which usually arises within the first few weeks of treatment (median time to onset, 15 days). Lorlatinib-induced hyperlipidemia is a common reason for temporary (but not permanent) discontinuation and dose-reduction of lorlatinib in the clinical setting, which necessitates the rapid initiation of lipid-lowering medications. Accordingly, the majority (>80%) of patients receive at least one lipid-lowering agent within 3 weeks of the first lorlatinib dose (median time to onset, 20 days), and ~25% of patients who received a lipid-lowering agent for hypercholesterolemia or hypertriglyceridemia require the addition of a second agent. While statins are the most commonly prescribed lipid-lowering agents to clinically manage lorlatinib-related hyperlipidemia, the choice and dosing of statins should be guided by the differential metabolism of the CYP450 pathway [16]. As lorlatinib is metabolized primarily by CYP3A4, the so-called 3A4 statin substrates atorvastatin, lovastatin, and simvastatin should not be co-administered as lipid-lowering agents in lorlatinib-treated patients. The non-3A4 statin substrates rosuvastatin, pravastatin, and pitavastatin are not significantly metabolized by CYP3A4 enzymes, and so the potential for adverse statin-lorlatinib interactions is notably reduced. Rosuvastatin is currently the only lipid-lowering agent recommended for lorlatinib-associated hyperlipidemia based on its low involvement with CYP450 enzymes that are able to interact with lorlatinib [17]. 

Given the improvements in prognosis and survival of patients with ALK translocations, a better understanding of the primary causes underlying lorlatinib-triggered hypercholesterolemia/hypertriglyceridemia might inform new therapeutic strategies to prevent or manage the undesirable lipid-modifying activity of lorlatinib in patients with ALK-positive NSCLC. Here, we tested the hypothesis that lorlatinib might directly promote hypercholesteremic and hypertriglyceridemic effects in human hepatocytes. First, using non-targeted lipidomics comprising ultrahigh pressure liquid chromatography accurate-mass quadrupole time-of-flight mass spectrometry with electrospray ionization (UHPLC-ESI-QTOF-MS/MS) [18], and imaging of neutral lipids, we explored the ability of lorlatinib to alter the lipidome of hepatoma tissue-derived Huh-7 and HepG2 cells [19,20,21,22,23,24], which were employed as substitutes for primary hepatocytes. Second, we investigated whether silibinin––a flavonolignan that functions as a hepatoprotectant in patients with acute and chronic liver injury [25,26,27,28,29,30]––might prevent the lipid-modifying activity of lorlatinib in hepatocytes. To predict potentially relevant drug–drug interactions if silibinin were used to clinically manage lorlatinib-associated hyperlipidemia, we finally explored the capacity of silibinin to interact with and block CYP3A4 activity in comparison with currently employed statins. We now provide first-in-class evidence that the hyperlipidemic effects of lorlatinib might involve, at least in part, the induction of an increased content of cholesterol and triglycerides in hepatic cells, which can be prevented or reversed by silibinin. 

## 2. Results

### 2.1. Hepatic Cells Treated with Lorlatinib Accumulate Cholesteryl Esters and Triglycerides

We first carried out MTT-based viability tests to evaluate the sensitivity of human hepatoma-derived Huh-7 cells, which closely mimic primary hepatocytes for drug metabolism and toxicity studies, to lorlatinib. As shown in Figure 1A, the IC_50_ of lorlatinib for Huh-7 cells (22 ± 5 μmol/L) was similar to that for ALK-negative A549 NSCLC cells (19 ± 5 μmol/L), but was tens or even hundreds of thousands higher than that obtained for H2228 and H3122 NSCLC cell lines (0.9 ± 0.2 and 0.003 ± 0.0002 μmol/L, respectively) harboring ALK rearrangements (variants 3a/b and 1, respectively) [31,32].

We then questioned whether lorlatinib used at concentrations efficacious against ALK-rearranged lung cancer cells but non-toxic to ALK-negative cells could promote substantial lipidome changes in Huh-7 cells (Figure 1B; Table 1). Accordingly, Huh-7 cells were treated with 1 μmol/L lorlatinib for 48 h, and the lipidome was characterized in a semi-targeted approach using a UHPLC-ESI-QTOF-MS/MS method, in which we matched mass spectra with commercial standards (SPLAH mix), databases such as METLIN-PCDL and Lipid MAPS (accessed on 1 July 2022), as well as corroboration of tentative lipid species upon obtaining their MS/MS spectrum. Representative chromatograms of lipidomic analyses in Huh-7 cells can be found in Appendix A.

We identified and quantified more than 100 lipid species belonging to six different families: diglycerides (DG), triglycerides (TG), phosphatidylcholines (PC), lysosophosphatidylcholines (LPC), phosphatidylethanolamines (PE), sphingomyelins (SM), and cholesterol esters (ChoE). Lorlatinib treatment was found to significantly increase several species of ChoE, numerous molecular species of long-chain TG, and several DG, LPC, and PC species (Figure 1C; Appendix A; Appendix A). 

### 2.2. Silibinin Fully Protects the Steady-State Lipidome of Hepatic Cells against the Hyperlipidemic Effects of Lorlatinib

Volcano plots and Venn diagrams of differentially accumulated lipid species revealed that, when used as a single agent, silibinin (100 μmol/L, 48 h) treatment significantly lowered the baseline abundance of many ChoE species in Huh-7 cells. Similarly, silibinin co-treatment completely blunted the significant lorlatinib-induced rise in numerous ChoE and TG/DG species in Huh-7 cells. Indeed, cells co-treated with silibinin and lorlatinib showed notable decreases in the abundance of various ChoE species, and this was accompanied by the compensatory generation of PC species (Figure 1C; Appendix A). Of note, no significant changes in any lipid species were observed when comparing silibinin-treated hepatocytes with those co-treated with silibinin and lorlatinib (Figure 1C; Appendix A).

Colorimetric enzymatic assays confirmed the ability of silibinin to inhibit the accumulation of cholesterol esters and triglycerides in lorlatinib-treated Huh-7 cells (Figure 1D).

### 2.3. Silibinin Prevents the Lorlatinib-Induced Chronic Accumulation of Neutral Lipids in Hepatic Cell

To evaluate the potential clinical relevance of silibinin’s ability to acutely suppress lorlatinib-induced hyperlipidemia in hepatic cells, we designed a protocol of repeated daily exposure to lorlatinib and/or silibinin using a silibinin concentration (10 μmol/L) that can be practically achieved in a clinical setting. We first employed a conventional index of fatty liver disease––the intracellular accumulation of neutral lipids (steatosis)––to confirm the ability of lorlatinib to trigger hypertriglyceridemia and hypercholesteremia. The results of Oil red O- and LipidTOX™-based staining of neutral lipids (cholesteryl esters and triglycerides) in lorlatinib-treated Huh-7 cells cultured with or without silibinin are shown in Figure 2A. As expected, in the lorlatinib only-treated arm, hepatic cells accumulated large amounts of lipid droplet-like neutral lipids, whereas this pattern was largely lost in the presence of clinically relevant concentrations of silibinin, which notably decreased the quantity and size of the lipid droplets formed by exposure to lorlatinib.

To confirm these findings, we employed an independent in vitro model of fatty liver disease involving non-tumorigenic HepG2 cells, which are routinely employed as an inducible fatty liver cell model to screen for therapeutic drugs to eliminate lipid accumulation. The results of Oil red O staining of lorlatinib-treated HepG2 cells cultured with or without silibinin are shown in Figure 2B. Lorlatinib-treated HepG2 cells contained a greater number of larger lipid droplets than untreated cells. By contrast, there were markedly fewer and smaller neutral lipid depots in the presence of silibinin even at the highest tested concentration of lorlatinib.

### 2.4. Silibinin Does Not Overlap the Binding Mode of Lorlatinib to Cytochrome P450 3A4 (CYP3A4) In Silico

To predict clinically relevant drug–drug interactions if silibinin were used to clinically manage lorlatinib-induced hyperlipidemic effects in hepatic cells, we explored the capacity of silibinin to interact with and block the lorlatinib-metabolizing CYP3A4 isoenzyme using in silico computational descriptions and in vitro biochemical assays. First, to shed some light on the capacity of silibinin to interact with CYP3A4, we performed a computational study comparing the predicted molecular interactions between silibinin and CYP3A4 with those of lorlatinib and ritonavir, which is the most potent CYP3A4A inhibitor in clinical use. As silibinin is almost a 1:1 mixture of the diastereomers A and B, we performed classical molecular dockings of silibinin A and B against the 3D crystal structure of human CYP3A4 bound to an inhibitor (PDB:7KVS) [33] (Figure 3).

To characterize the potential binding of silibinin A and B to the large catalytic cavity of CYP3A4, which allows CYP3A4 to bind a broad range of substrates as compared with other CYP isoenzymes, a total of 75 flexible docking runs were set around the putative binding sites of lorlatinib and ritonavir using AutoDock Vina implemented by YASARA. The resulting binding energies with the docking simulations of silibinin A (−10.323 kcal/mol) and B (−9.561 kcal/mol) were similar to those obtained with lorlatinib (−9.876 kcal/mol) and ritonavir (−9.962 kcal/mol). However, closer inspection of the different conformations revealed that the predicted interaction of silibinin A and B with the active site cavity of CYP3A4 shared only 39–46% of the 12 amino acid residues involved in the lorlatinib binding mode, and 45–59% of the 22 participating amino acids involved in the ritonavir binding mode (Table 2).

Unlike lorlatinib and ritonavir, neither silibinin A nor silibinin B were predicted to interact with PHE304, which is critically important for CYP3A4 substrate specificity and catalytic capacity [34,35,36]. Moreover, neither silibinin A nor silibinin B were predicted to interact with a majority of the hydrophobic pocket near the I-helix (the so-called Phe-1 site) including PHE108, LEU210, LEU211, PHE241, ILE301, and PHE304 [37,38] (Figure 3). Silibinin is, therefore, predicted to interact with the catalytic site of CYP3A4 through a unique binding mode lacking the involvement of the gatekeeper residues regulating the access of CYP3A4 substrates to the heme group, suggesting a highly improbable interaction between silibinin and lorlatinib at the CYP3A4 active site. 

### 2.5. Silibinin Is a Weak Inhibitor of the Lorlatinib-Metabolizing Cytochrome P450 3A4 (CYP3A4) Isoenzyme 

The CYP3A4 inhibition propensity of silibinin in comparison with that of recommended (non-3A4 substrates such as rosuvastatin) and non-recommended (3A4 substrates such as simvastatin) statins to clinically manage lorlatinib-associated dyslipidemia was initially predicted using the DL-CYP prediction server [39], a free web tool to evaluate the tendency of small molecules to inhibit five major CYP450 isoforms (1A2, 3C9, 2C19, 2D6, and 3A4). Using this deep autoencoder multitask neural network trained on more than 13,000 compounds, simvastatin showed the highest predicted value (0.93), which ranked very close to that obtained with the positive control ritonavir (0.97) (Table 3). The tendency of CYP3A4 inhibition by rosuvastatin was predicted to be as low as 0.2, whereas silibinin had no predicted inhibitory effect in silico (0.06) (Table 3).

To facilitate the understanding of the predicted tendencies, we performed MD simulations for each of the CYP3A4-drug complexes. This approach considers the protein flexibility at the target-binding site during the molecular recognition process, allowing the user to confirm the kinetic stability and validate the binding poses obtained by docking. The CYP3A4 protein backbone root mean square deviation (RMSD) plots of the drugs’ heavy atoms, measured after superimposing CYP3A4 onto its reference structure during the MD simulation, were prepared in parallel. The approach is summarized in Figure 4 and shows as follows: the best poses of ritonavir, silibinin A, silibinin B, simvastatin, and rosuvastatin coupled to the catalytic cavity of CYP3A4 before (0 ns) and after (100 ns) the MD simulation, the time evolution of RMSD relative to the initial structure of CYP3A4 in the absence and presence of drugs, the binding free energy calculations under the MM/PBSA approximation from the entire MD simulation trajectory of 100 ns (or last 30 ns), and the identification of amino acid residues participating in the drug-CYP3A4 binding pocket. The MM/PBSA parameters, which estimate the free energy of the binding of small ligands to biological macromolecules, allows the correct ranking of drug candidates based on the fact that more positive energies indicate the occurrence of stronger binders (e.g., “strong” > 25 kcal/mol versus “weak” < 25 kcal/mol). Likewise, the results predicted a high affinity of ritonavir and simvastatin for the catalytic site of CYP3A4, reaching MM/PBSA values of ~70 and 40 kcal/mol, respectively (Figure 4). Silibinin A mimicked rosuvastatin as a predicted weak inhibitor of CYP3A4, showing MM/PBSA values of <20 kcal/mol. Silibinin B was predicted to encounter steric hindrance at the catalytic activity of CYP3A4 given its notable −60 kcal/mol MM/PBSA value (Figure 4). 

The inhibitory effects of graded concentrations of simvastatin, rosuvastatin, silibinin, and Eurosil 85^®^/Euromed –a standardized pharmaceutical preparation containing 60% silibinin with an enhanced bioavailability (>80%) that is commonly used in clinical research [40,41]—on CYP3A4 activity were finally evaluated using the Vivid^®^ CYP450 Red screening assay, which employs microsomes prepared from insect cells (baculosomes) expressing a human CYP450 isoenzyme (CYP3A4 in this case) and human cytochrome P450 reductase, and utilizing the benzyloxy–methyl–resofurin (BOMR) red substrate. The Vivid^®^ BOMR substrate is a blocked dye that yields a minimal fluorescence signal until metabolized by CYP3A4 into products that are highly fluorescent in aqueous solutions (Figure 5, *left panel*). Simvastatin strongly inhibited the enzymatic activity of CYP3A4, exhibiting an IC_50_ value as low as 1.8 ± 0.9 μmol/L. Up to 10-fold higher concentrations of silibinin (IC_50_ = 16.7 ± 8.9 μmol/L) and Eurosil 85^®^/Euromed (12.1 ± 1.2 μmol/L) were necessary to achieve the half-inhibitory concentration of CYP3A4 activity (Figure 5, *right panel*). Rosuvastatin likewise failed to show any significant inhibitory effect on CYP3A4 activity even when tested at very high concentrations (>300 μmol/L).

## 3. Discussion

Elevated plasma levels of cholesterol and triglycerides are a common adverse effect in the majority of patients with ALK-positive NSCLC treated with lorlatinib [14,15,17]. The molecular mechanisms of lorlatinib-related lipid disorders are likely multifactorial and are currently unclear. Because hyperlipidemia is a classic feature of nephrotic syndrome, the fact that lorlatinib is apparently the only ALK-TKI reported to induce acute kidney injury involving renal cyst formation or progression suggests that minimal change disease (nephrotic syndrome) should be viewed as a possible secondary cause of lorlatinib-induced hyperlipidemia [42,43,44,45]. Here, we provide pre-clinical evidence that the elevated plasma cholesterol and triglyceride levels in lorlatinib-treated lung cancer patients might be due, at least in part, to direct alterations in the hepatic accumulation of lipid intermediates. Moreover, we reveal that the flavonolignan silibinin has the capacity to protect hepatic cells against the lipid-modifying activity of lorlatinib (Figure 6).

Lorlatinib is rapidly absorbed to achieve peak plasma concentrations of >2 μmol/L 0.5–4 h after a single 100 mg oral dose and is widely distributed in different tissues [46,47], with the highest concentration found in the liver [48]. The hepatic accumulation of lorlatinib can be explained in terms of its low metabolism in this organ, with only 12% of lorlatinib metabolized during first-pass hepatic metabolism or pre-systemic metabolism. We show that acute exposure to a clinically relevant concentration of lorlatinib (1 μmol/L) promotes a rapid and significant buildup of cholesteryl esters and tri-/di-glycerides in hepatic cells. We also found that longer exposure to lorlatinib drives the hallmarks of hepatic steatosis, such as the abnormal and excessive accumulation of neutral lipids in liver cells. While our study was not designed to investigate the precise molecular mechanisms involved in the hepatic alterations in cholesterol and triglycerides homeostasis in response to lorlatinib, it is reasonable to suggest that there is overlap in the targeted pathways for ALK-rearranged NSCLC growth and those that regulate hepatic function, leading to on-target or closely related off-target hepatic toxicity. It will be critical to understand the mechanistic pathways by which lorlatinib induces a dysregulation of lipid and cholesterol metabolism in hepatic cells. Because one of the major mechanisms of drug-induced liver injury (including TKIs such as lorlatinib) centrally involves mitochondrial perturbation and dysfunction [49,50], mitochondria might be viewed as a probable target through which lorlatinib alters lipid metabolism in human hepatocytes. Particularly, drugs inducing impairment of fatty acids oxidation are known to lead to lipid accumulation (steatosis) and steatohepatitis [50], thus suggesting that a plausible hypothesis for the hyperlipidemic mechanism of action of lorlatinib might involve a decrease in lipid catabolism in response to TKI-related mitochondrial dysfunction in hepatic cells, leading to a disbalance between fatty acid *β*-oxidation activity and cholesteryl esters/triglycerides synthesis [51]. Future studies will be needed to address this possibility. We acknowledge, however, that our pre-clinical findings in cell culture do not preclude the possibility that other mechanisms involving changes in peripheral tissues (e.g., adipose tissue, muscle), lipid uptake by the liver, or intestinal lipid absorption can contribute to the lorlatinib-induced lipid metabolic effects. 

The flavonolignan silibinin has been shown to significantly temper the accumulation of hepatic and biliary triglycerides and cholesterol in cultured cells and animal models, and in patients with type 2 diabetes and chronic liver disease. We show here that silibinin suppresses the lipid-modifying activity of lorlatinib in human hepatic cells. Regardless of the mechanism through which silibinin protects the lipidome of hepatic cells against lorlatinib (e.g., by increasing the shift of fatty acids from triglycerides towards phospholipids and/or increasing the endogenous cholesterol conversion to bile acids), silibinin might represent an idoneous lipid-lowering agent in new treatment regimens for patients with NSCLC at the highest risk of developing brain metastases, such as those continuously exposed to ALK-TKIs [52]. As lorlatinib is effective against central nervous system metastasis even in patients pretreated with first- and second-generation ALK-TKIs [10], the known capacity of silibinin to provide better control of brain disease and allow a higher proportion of patients with NSCLC to receive additional lines of treatment [53,54] could be exploited by utilizing its ability to ameliorate lorlatinib-induced hyperlipidemia. 

The low hepatic extraction ratio (<30%) of “low” clearance drugs such as lorlatinib can drive a change in the plasma counterpart when co-administered with another drug that is a metabolic inhibitor or inducer. Because lorlatinib is metabolized primarily by the CYP3A4 isoenzyme [55,56], lipid-lowering agents that do not interact with CYP3A4, such as rosuvastatin, should be co-administered in lorlatinib-treated patients. Our multi-faceted approach involving computational modeling, neural network-based prediction models to classify CYP inhibition, and in vitro validation using recombinant human CYP3A4, confirms and further extends the notion that exposure to standard doses of silibinin-containing milk thistle preparations leading to relatively low peak systemic concentrations (1–10 μmol/L) are expected to be accompanied by low-risk for an inhibitory interaction of the first-pass clearance of lorlatinib at the hepatic level [57]. Nonetheless, because notably higher intestinal concentrations of silibinin can be achieved when using gram doses of certain silibinin formulations that have been tested in patient populations [57,58], it cannot be excluded that clinically relevant inhibitory interactions with the lorlatinib-metabolizing CYP3A4 isoenzyme might occur at the intestinal level. Indeed, a limitation of our study is that the capacity of silibinin to protect hepatocytes from the hyperlipidemic activity of lorlatinib was observed in two hepatoma-derived cell lines (Huh-7 and HepG2) that naturally express low expression of drug metabolizing enzymes such as CYP3A4 [59,60,61]. We therefore acknowledge that our cell-free evaluations of the in silico and in vitro metabolic interactions between lorlatinib and silibinin (versus statins) using computational descriptions (with crystal structures of CYP3A4) and biochemical assays (with recombinant CYP3A4) cannot be mechanistically extrapolated to the cell-based experiments showing the ability of silibinin to suppress the hypercholesterolemic and hypertriglyceridemic effects of lorlatinib in Huh-7 and HepG2 cells. We failed to observed significant toxic effects when combining lorlatinib with silibinin concentrations as high as 100 μmol/L to carry out lipidomic studies with Huh-7 cells growing confluent for more than 1 month, thereby ensuring an increased expression and catalytic activity of CYP3A4 [62,63]. However, the extrapolation of the in vitro results to accurately predict the toxicity as it would occur in vivo might require the usage of the newly developed human hepatoma cell lines such as HepaRG [64,65,66], which shows stable expression of liver-specific functions over a long period in culture, including high levels of CYP3A4. 

## 4. Materials and Methods

### 4.1. Materials 

Silibinin was purchased from Sigma-Aldrich (St. Louis, MO, USA; Cat. #S0417). Simvastatin (MK 733) and rosuvastatin (ZD4522) were obtained from Selleck Chemicals LLC (Houston, TX, USA; Cat. #S1792 and S2169, respectively). Eurosil^85^/Euromed was kindly provided by Meda Pharma S.L. (Barcelona, Spain). All reagents were dissolved in sterile dimethylsulfoxide to prepare 10 mmol/L stock solutions, which were stored in aliquots at −20°C until use. Working concentrations were diluted in culture medium prior each experiment.

Grade MS methanol (MeOH) and 2-propanol, ammonium formate, formic acid, acetic acid, methyl *tert*-buthyl ether (MTBE) and standards for calibration curves were purchased from Sigma-Aldrich. Water (Milli-Q grade) was obtained from a Milli-Q integral water purification system (Millipore Corp., Burlington, MA, USA). For internal standards, SPLASH Lipidomix was purchased from Avanti Polar lipids (Alabaster, AL, USA; Cat. #330707).

### 4.2. Cell Lines

Huh-7 and HepG2 cells were kindly provided by Dr. Jose Manuel Fernández-Real in our institution (IDIBGI), and were cultured in complete Dulbecco′s modified Eagle′s medium (Gibco/Invitrogen, Carlsbad, CA, USA) supplemented with 10% fetal bovine serum, 2 mmol/L L-glutamine, and 100 IU/mL penicillin-streptomycin (all from Gibco/Invitrogen, Carlsbad, CA, USA).

### 4.3. Cell Viability Assays

Cell viability effects of lorlatinib were determined using the colorimetric MTT (3-4,5-dimethylthiazol-2-yl-2,5-diphenyl-tetrazolium bromide) reduction assay. Dose–response curves to graded concentrations of lorlatinib were plotted as a percentage of the control cell absorbance, which was obtained from control cells containing the vehicle (DMSO *v/v*) processed simultaneously. For each treatment, cell viability was evaluated using the following equation: (OD_570_ of the treated sample/OD_570_ of the untreated sample) × 100. Sensitivity to agents was expressed in terms of the concentrations required for a 50% (IC_50_) reduction in cell viability. Since the percentage of control absorbance was considered to be the surviving fraction of cells, the IC_50_ values were defined as the concentration of drug that produced 50% reduction in control absorbance (by interpolation).

### 4.4. Non-Targeted Lipidomics

Lipid extraction, preparation of external calibration curves, and quality control approaches were carried out as described [18]. Samples (2 μL) were injected directly into a 1290 Infinity UHPLC system coupled with a dual Agilent jet stream ESI source to a 6550 QTOF-MS instrument (Agilent Technologies, Santa Clara, CA, USA). UHPLC-ESI-QTOF-MS conditions, data analysis and indirect quantification of selected lipid species using MassHunter Qualitative Analysis B.07.00 software (Agilent Technologies), and corroboration of the tentative characterization of compound match entities using the Lipid Maps database (www.lipidmaps.org, accessed on 1 July 2022) and targeted MS/MS acquisition on LC-QTOF-MS instrument, have been previously described in detail [18].

Differentially expressed lipid species from untargeted lipidomic data were analyzed by generating volcano plots indicating statistical significance versus magnitude of fold-change for each lipid species (cutoff of Benjamini–Hochberg corrected *p*-value < 0.05 and log2 fold-change > 1 or <1). 

### 4.5. Triglyceride and Cholesterol Quantification

Quantification of cholesteryl esters and triglycerides in Huh-7 cells cultured in the absence/presence of lorlatinib and/or silibinin was performed using cholesterol (Cat. #MAK043) and triglyceride (Cat. #MAK266) quantification kits (Sigma-Aldrich). 

### 4.6. Accumulation of Neutral Lipids 

For assessment of neutral lipid formation, cells were washed in PBS, fixed with 4% paraformaldehyde for 30 min, washed in distilled water, and then briefly incubated in 60% isopropanol. Following equilibration, cells were stained with a working solution (60%) of Oil Red O for 20 min. An Oil Red O stock solution was prepared by dissolving 0.35% *w/v* Oil Red O in 100 mL 100% isopropanol. HCS LipidTOX™ Green Neutral Lipid Stain (Invitrogen; Cat. #H34475) was employed as a second staining cocktail for visualization of intracellular neutral lipids. Formaldehyde fixation and preparing/using the labeling solution was carried out as per the manufacturer’s instructions. 

### 4.7. Prediction of CYP450 Inhibition In Silico

The CYP450 inhibition propensity of silibinin, simvastatin, and rosuvastatin was predicted with the DL-CYP Prediction Server (http://www.pkumdl.cn:8000/deepcyp/home.php, accessed on 1 July 2022), a free web tool to evaluate the tendency of small molecules to inhibit five CYP450 major isoforms, namely 1A2, 2C9, 2C19, 2D6, and 3A4 [39]. All structure-data files (sdf) for tested compounds were downloaded from PubChem (National Center for Biotechnology Information) and used as the inputs to predict their level of inhibition against human CYP450 isoforms. The CYP3A4 inhibitor ritonavir was used as a positive control to predict the inhibition of CYP3A4 activity. The results were expressed as values between 0 and 1. 

### 4.8. Docking Calculations, Molecular Dynamics Simulations, and Binding Free Energy Analysis

Docking calculations, molecular dynamics (MD) simulations, and molecular mechanics Poisson–Boltzmann surface area (MM/PBSA) calculations to determine the alchemical binding free energy of silibinin, simvastatin, and rosuvastatin against CYP3A4 7KVS were performed using procedures described in previous works from our group [67,68,69,70,71,72,73]. To perform the docking studies with AutoDockVina (v1.1.2, San Diego, CA, USA), the CYP3A4 7KVS crystal structure was transformed to the PDBQT format, including the atomic charges and atom-type definitions. These preparations were performed using the AutoDock/Vina plugin with scripts from the AutoDock Tools package [74]. YASARA dynamics v19.9.17 (Vienna, Austria) was employed for all MD simulations with the AMBER14 force field. All simulation steps were run using a pre-installed macro (md_run.mcr) within the YASARA suite. Data were collected every 100 ps during 100 ns. The MM/PBSA calculations of solvation binding energy were calculated using the YASARA macro md_analyzebindenergy.mcr, with more negative values indicating instability. MM/PBSA was implemented with the YASARA macro md_analyzebindenergy.mcr to calculate the binding free energy with solvation of the ligand, complex, and free protein, as previously described [67,68,69,70,71,72,73]. All of the figures were prepared using PyMol 2.0 software and all interactions were detected using the protein-ligand interaction profiler (PLIP) algorithm [75].

### 4.9. CYP3A4 Inhibition

The characterization of the CYP3A4 inhibitory potency of silibinin, simvastatin, and rosuvastatin was outsourced to the SelectScreen P450 Profiling Service (ThermoFisher Scientific, Waltham, MA, USA) using CYP3A4-transfected baculosomes from the Vivid^®^ CYP450 Red assay platform (Cat. No. P2856). The stringent validation process ensures the highest quality data possible for the 10-point titration, which was performed in duplicate (*n* = 3). An additional two control wells for each compound concentration tested are conducted to detect autofluorescent compound interference. Strict quality control protocols ensure that any assay results not meeting set specifications will be automatically repeated, this includes performing IC_50_ assays with a known inhibitor for each P450 on a per plate basis to ensure the validity of the data. To calculate percent inhibition due to presence of test compounds (i.e., silibinin, simvastatin, and rosuvastatin) or positive inhibition (i.e., ketoconazole for CYP3A4, which enables the subtraction of background during data analysis) we used the following equation:$$\%\;\mathrm{inhibition}=(1\;–\;\frac{X–B}{A–B})\times 100\%$$
where X is the fluorescence intensity of test compound, A is the fluorescence intensity observed in the absence of inhibitor (DMSO-only control that accounts for possible inhibition caused by introduction of test compounds originally dissolved in organic solvents), and B is the fluorescence intensity observed in the presence of positive inhibition control. 

### 4.10. Statistical Analysis

All cell-based observations were confirmed by at least three independent experiments performed in triplicate for each cell line and for each condition. Data are presented as mean ± SD. Two-group comparisons were performed using Student′s *t*-test for paired and unpaired values. Comparisons of means of ≥3 groups were performed by ANOVA, and the existence of individual differences, in case of significant F values with ANOVA, was tested by Scheffé′s multiple contrasts: *p*-values < 0.01 and <0.001 were considered to be statistically significant (denoted as * and **, respectively). All statistical tests were two-sided.

## 5. Conclusions

Hyperlipidemia, which encompasses hypercholesteremia and hypertriglyceridemia, is a unique adverse side effect of lorlatinib that is mainly controlled with dose interruptions/modification and lipid-lowering therapies. We report here that clinically relevant concentrations of lorlatinib directly modify the lipid profile of human hepatocytes. Silibinin, a flavonolignan with remarkable clinical efficacy against brain metastases in patients with ALK-rearranged NSCLC and capable of overcoming resistance to ALK-TKIs in vitro [76,77], protects the native lipidome of hepatic cells against the hyperlipidemic effects of lorlatinib and prevents lipid accumulation at therapeutically relevant concentrations. Although silibinin might become a new candidate to clinically manage the undesirable lorlatinib-induced hypercholesterolemic and hyperlipidemic effects in human hepatic cells, further studies will be required to fully define the inhibitory potential of silibinin against the lorlatinib-metabolizing CYP3A4 isoenzyme when used as specific orally bioavailable formulations (e.g., Eurosil 85^®^) before suggesting its therapeutic combination with lorlatinib in a clinical setting with ALK-rearranged lung cancer patients.

## Figures and Tables

**Figure 1 ijms-23-09986-f001:**
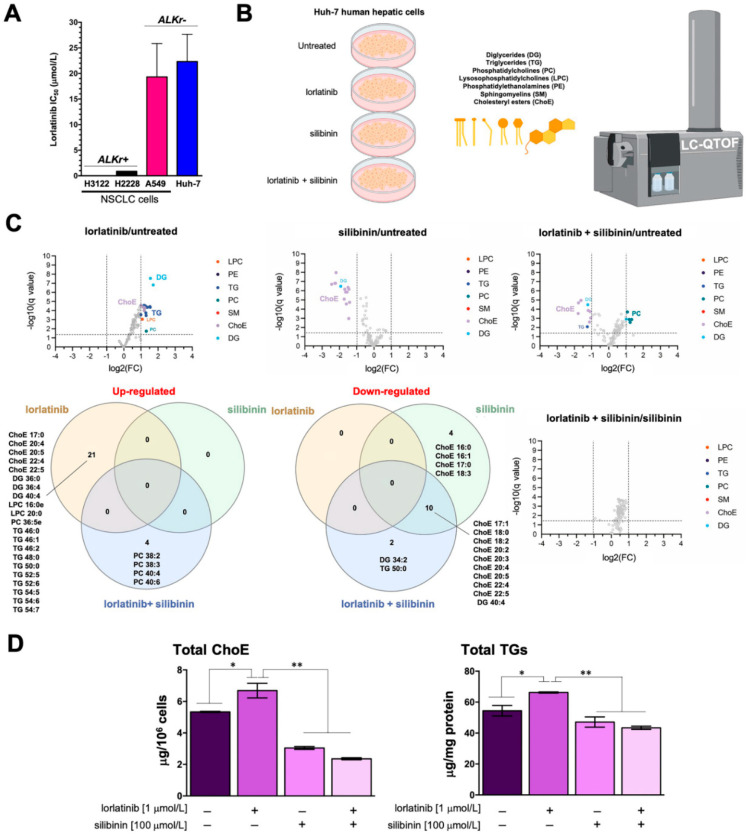
**Silibinin prevents lorlatinib-driven lipidome alterations in human hepatic cells:** (**A**) bar graphs of the IC_50_ values for each cell line calculated from the MTT assays as described in “Materials and Methods”; the results are presented as the means (*columns*) ± S.D. (*bars*) (*n* ≥ 3 in technical replicates); (**B**) to test the hypothesis that lorlatinib could directly promote the accumulation of triglycerides and/or cholesterol in human hepatocytes, we carried out an untargeted UHPLC-ESI-QTOF-MS/MS-based lipidomic analysis of > 100 molecular lipid species in Huh-7 cells cultured in the absence/presence of lorlatinib and/or silibinin; (**C**) Volcano plots and Venn diagrams of the results from lipidomic analyses (**B**) in human Huh-7 cells treated with lorlatinib (1 μmol/L), silibinin (100 μmol/L) or their combination for 48 h. Each dot represents a lipid species with its corresponding mean Log2 fold-change (FC) (*x* axis) and Benjamini–Hochberg corrected *p*-value (-log10, *y* axis). Colored dots illustrate differential lipid species, using a cutoff of *p* < 0.05 and log2FC > 1 or <1. (**D**). ELISA-based quantification of cholesteryl esters (*left*) and triglycerides (*right*) in Huh-7 cells treated with lorlatinib (1 μmol/L), silibinin (100 μmol/L) or their combination for 48 h. *p*-values < 0.01 (*) and <0.001 (**).

**Figure 2 ijms-23-09986-f002:**
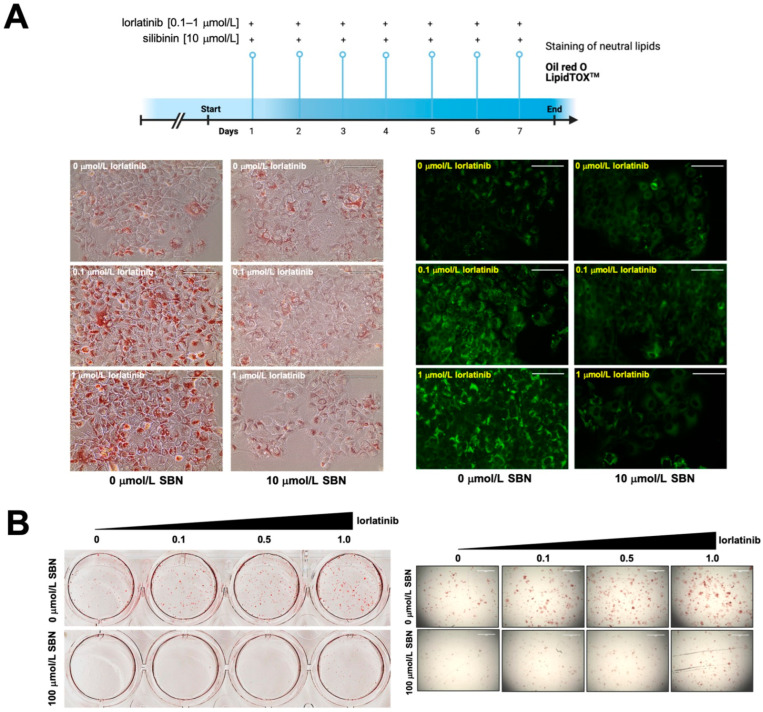
**Silibinin inhibits the chronic accumulation of neutral lipids in lorlatinib-treated hepatic cells** (**A**) Conventional indices of fatty acid liver disease ––lipid droplet content in hepatic cells––were assessed using Oil Red O- and LipidTOX green-based staining of neutral lipids in Huh-7 (**A**) and HepG2 (**B**) cells chronically (**A**) or acutely (**B**) exposed to lorlatinib in the absence or presence of silibinin. (Chronic exposure: 7 days at 10 μmol/L silibinin; acute exposure: 48 h at 100 μmol/L silibinin). Scale bar = 100 μm.

**Figure 3 ijms-23-09986-f003:**
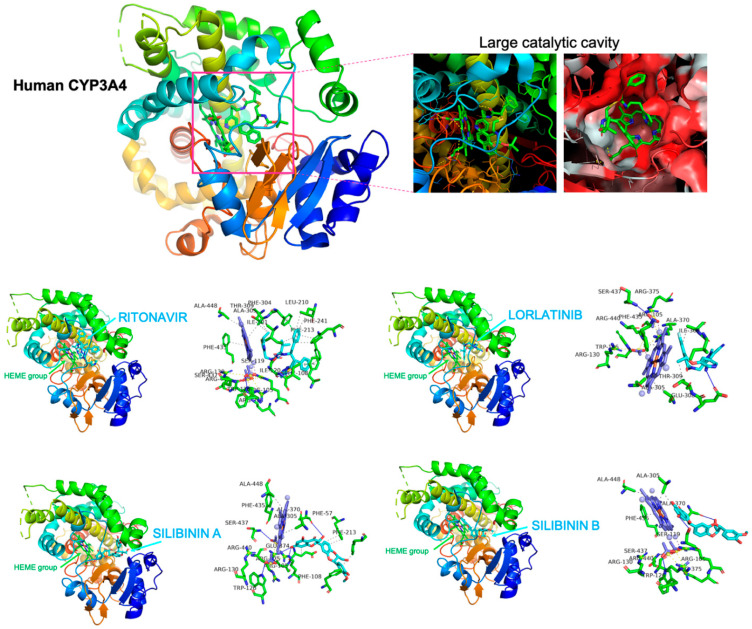
**Silibinin does not share the binding mode of lorlatinib to CYP3A4.** Figure depicts the backbone of the overall crystal structure of CYP3A4 (7KVS) with rainbow colors showing the best docked poses of silibinin A, silibinin B, ritonavir, and lorlatinib at the catalytic site. Each inset shows the detailed interactions of silibinin docked to CYP3A4, indicating the participating amino acids involved in the interaction and the type of interaction (hydrogen bonds, hydrophilic interactions, salt bridges, II-stacking, etc).

**Figure 4 ijms-23-09986-f004:**
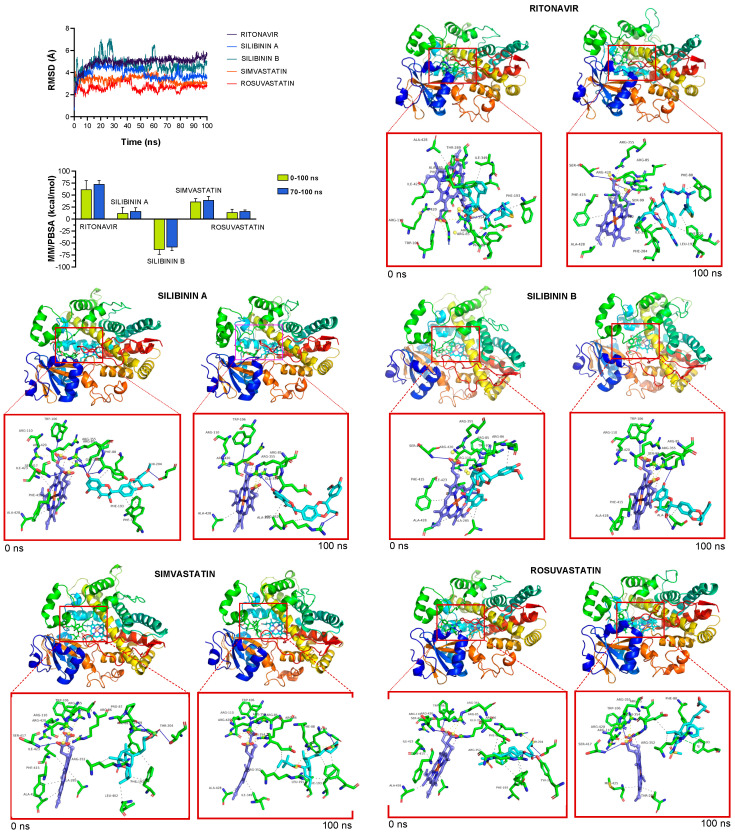
**Incorporation models of CYP3A4-targeting drugs: ritonavir and statins *vs* silibinin.** The root mean square deviation (RMSD, Å) of each drug’s heavy atoms over the simulation time, measured after superposing the protein onto its reference structure, and the molecular mechanics Poisson–Boltzmann surface area (MM/PBSA) binding energy analyses calculated from the entire trajectory of the 100 ns (or last 30 ns) MD simulation, are shown. The best poses of ritonavir, silibinin A, silibinin B, simvastatin, and rosuvastatin coupled to the catalytic site of CYP3A4 before (0 ns) and after (100 ns) the molecular dynamics (MD) simulation are shown. The protein is represented as a function of the hydrophobicity of its surface amino acids and the Na^+^ and Cl^−^ ions have been eliminated to facilitate visualization. Each inset shows the detailed interactions of the participating amino acids involved and the type of interaction (hydrogen bonds, hydrophilic interactions, salt bridges, Π-stacking, etc.).

**Figure 5 ijms-23-09986-f005:**
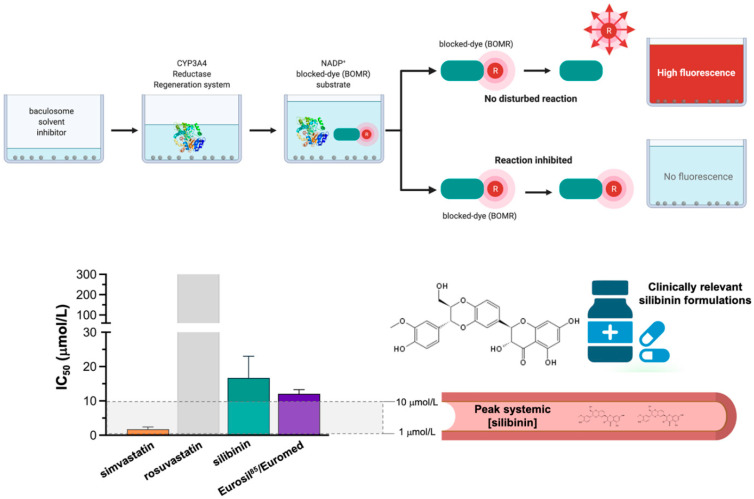
**Potency of human CYP3A4 inhibition: statins *vs* silibinin.** Percent inhibition and IC_50_ values of simvastatin, rosuvastatin, silibinin, and Eurosil 85^®^ against recombinant human CYP3A4 expressed in baculosomes using the Vivid BOMR Red substrate. The results are presented as the means (*columns*) ± S.D. (*bars*) (*n* = 3 experimental curves in technical duplicates).

**Figure 6 ijms-23-09986-f006:**
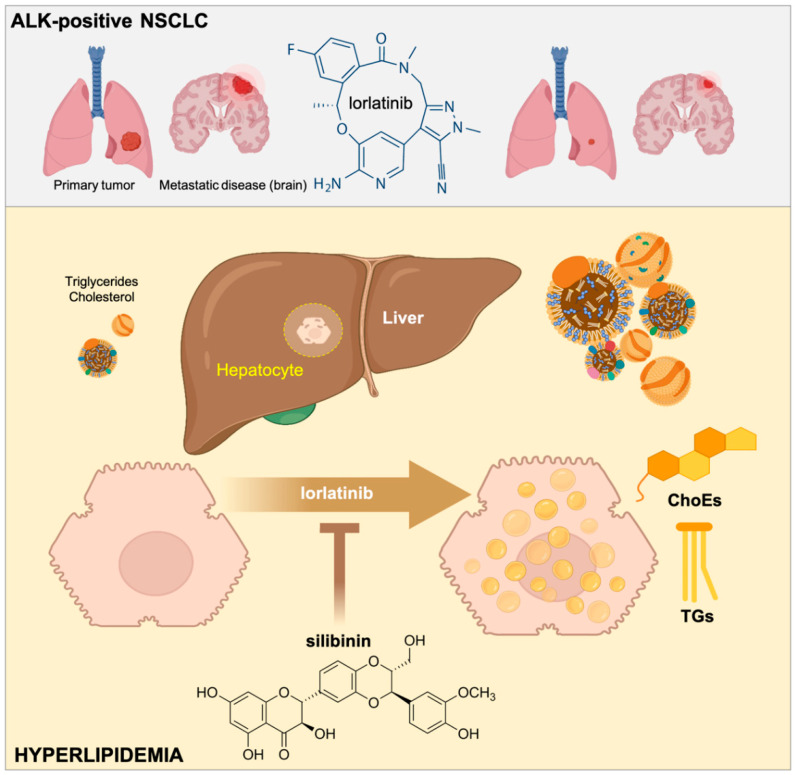
**Silibinin suppresses the hyperlipidemic effects of lorlatinib in hepatic cells.** The third generation ALK-TKI lorlatinib promotes the accumulation of cholesterol and triglycerides in human hepatic cells. The flavonolignan silibinin protects hepatic cells against the hypertriglyceridemic and hypercholesteremic effects of lorlatinib. Silibinin is a new candidate to clinically manage the undesirable hyperlipidemic activity of lorlatinib in patients with ALK-rearranged lung cancer.

**Table 1 ijms-23-09986-t001:** Lipid species (*n* = 124) identified using a UHPLC-ESI-QTOF-MS/MS method (e:ester).

DG (*n* = 9)	TG (*n* = 25)	PC (*n* = 40)	LPC (*n* = 9)	PE (*n* = 4)	SM (*n* = 24)	ChoE (*n* = 13)
DG 34:1	TG 46:0	PC 30:0	LPC 15:0	PE 32:0	SM 32:0	ChoE 16:0
DG 34:2	TG 46:1	PC 31:0	LPC 16:0	PE 36:4	SM 32:1	ChoE 16:1
DG 34:3	TG 46:2	PC 32:0	LPC 16:0e	PE 36:5e	SM 32:2	ChoE 17:0
DG 36:0	TG 48:0	PC 32:1	LPC 18:0	PE 38:5e	SM 33:1	ChoE 17:1
DG 36:1	TG 48:1	PC 32:1e	LPC 18:0e		SM 34:1	ChoE 18:0
DG 36:2	TG 48:2	PC 32:2	LPC 18:1		SM 34:2	ChoE 18:2
DG 36:3	TG 48:3	PC 33:0	LPC 18:2		SM 35:0	ChoE 18:3
DG 36:4	TG 50:0	PC 33:1	LPC 20:0		SM 35:1	ChoE 20:2
DG 40:4	TG 50:1	PC 33:2	LPC 20:3		SM 36:0	ChoE 20:3
	TG 50:2	PC 34:0			SM 36:1	ChoE 20:4
	TG 50:3	PC 34:1			SM 36:2	ChoE 20:5
	TG 50:4	PC 34:1e			SM 38:1	ChoE 22:4
	TG 51:2	PC 34:2			SM 38:2	ChoE 22:5
	TG 52:1	PC 34:2e			SM 39:1	
	TG 52:2	PC 34:3			SM 40:0	
	TG 52:3	PC 34:4			SM 40:1	
	TG 52:4	PC 35:1			SM 40:2	
	TG 52:5	PC 35:2			SM 41:1	
	TG 52:6	PC 35:4			SM 41:2	
	TG 54:2	PC 36:0			SM 42:1	
	TG 54:3	PC 36:1			SM 42:2	
	TG 54:4	PC 36:2			SM 42:3	
	TG 54:5	PC 36:2e			SM 43:1	
	TG 54:6	PC 36:3			SM 43:2	
	TG 54:7	PC 36:4				
		PC 36:5				
		PC 36:5e				
		PC 38:2				
		PC 38:3				
		PC 38:4				
		PC 38:5				
		PC 38:5e				
		PC 38:6				
		PC 38:6e				
		PC 40:4				
		PC 40:4e				
		PC 40:5				
		PC 40:6				
		PC 42:4e				
		PC 42:5e				

**Table 2 ijms-23-09986-t002:** Details of the interaction between silibinin A/B, ritonavir and lorlatinib to CYP3A4.

ΔG (kcal/mol)	K_d_ [nM]	Drug	Residues Involved in the Interaction (7KVS.pdb)
−10.323	27.1	silibinin A	TYR 53, PHE 57, ASP 76, ARG 105, ARG 106, PHE 108, GLY 109, PHE 213, PHE 215, PHE 220, ILE 223, THR 224, PRO 227, ILE 230, VAL 240, ALA 370, MET 371, ARG 372, LEU 373, GLU 374, ARG 375, **HEME 601**
−9.962	49.9	ritonavir	PHE 57, ARG 105, ARG 106, PHE 108, MET 114, SER 119, ILE 120, LEU 210, LEU 211, PHE 213, PHE 241, ILE 300, ILE 301, PHE 304, ALA 305, THR 309, ILE 369, ALA 370, MET 371, ARG 372, LEU 373, GLU 374, **HEME 601**
−9.876	57.6	lorlatinib	PHE 57, ARG 105, SER 119, LEU 211, PHE 304, GLU 308, THR 309, SER 312, ILE 369, ALA 370, MET 371, LEU 373, LEU 482, **HEME 601**
−9.651	84.3	silibinin B	PHE 57, ARG 105, ARG 106, PRO 107, PHE 108, SER 119, ILE 301, ALA 305, THR 309, ALA 370, MET 371, ARG 372, LEU 373, GLU 374, **HEME 601**

**Table 3 ijms-23-09986-t003:** Prediction of human cytochrome P450 inhibition.

	Cytochrome P450 Isoforms				
Drug	1A2	2C9	2C19	2D6	3A4
Ritonavir	0.00	0.34	0.36	0.01	0.97
Simvastatin	0.00	0.02	0.04	0.00	0.93
Rosuvastatin	0.00	0.45	0.18	0.00	0.2
Silibinin	0.00	0.02	0.04	0.00	0.06

## Data Availability

All data generated or analyzed during this study are included in this published article (and its Appendix A.

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
