# Peer review of "Silibinin Suppresses the Hyperlipidemic Effects of the ALK-Tyrosine Kinase Inhibitor Lorlatinib in Hepatic Cells"

_ijms, 2022, doi:10.3390/ijms23179986_

Round 1

Reviewer 1 Report

In this manuscript, Verdura et al. investigated the effects of silibinin on the suppression of hyperlipidemia induced by lorlatinib in hepatic cell lines. The significance of authors’ findings is limited due to a lack of solid evidence to draw their conclusions. The data presentation also needs to be improved.

Major comments are as follows.

There are some misuses of terminology, e.g., cell-autonomous and reprogram. To be sure of the phenomenon of “cell-autonomously”, the authors need to specify cell type and provide evidence in primary cells, e.g. hepatocytes. To provide in vivo data, the authors need to include both ALK null mice and CYP3A4 null mice. Cell lines are transformed and immortalized, and thereby they are not identical to primary hepatocytes whatsoever in terms of gene mutation and expression. There is no such thing of cell autonomous event in cell line culture unless a 3-D culture with multiple cell types, including wild-type and gene knockout cells, is to be used. Mouse chimera experiments, e.g., using irradiated mice plus bone marrow transplantation, must be performed to demonstrate a cell autonomous effect. In terms of “reprogramming”, this phenomenon is usually regulated epigenetically and it is irreversible. What the authors demonstrated was simply an “alteration” in lipid profiles in cell lines.

In Fig. 1, the authors should list the more than 100 lipid species included in the lipidomic analysis in Materials and Methods. To be more informative, the Venn diagram should be used to illustrate the results of comparisons among 4 groups shown in Fig. 1C and Fig. 1D. 

In Fig. 3 and Fig. 4, the data/findings were MISSING. Only legends were seen.

In Fig. 5, a negative or irrelevant control of CYP450 enzyme should be included to demonstrate specificity of CYP3A4.

Reviewer 2 Report

Hyperlipidemia, which includes hypercholesterolemia and hypertriglyceridemia, is a unique undesirable side effect of lorlatinib that is mainly controlled by dose interruption / modification and lipid-lowering therapies. The authors of this study prove that silibinin, a flavonolignan, protects the native lipids of liver cells against the hyperlipidemic effects of lorlatinib and prevents lipid accumulation at therapeutically relevant concentrations. Although silibinin may become a new candidate for the clinical treatment of undesirable cellular (primary) hyperlipidemia induced by lorlatinib in human liver cells, further studies are needed to fully determine the inhibitory potential of silibinin to the CYP3A4 isoenzyme that metabolizes lorlatinib and the final therapeutic application of a combination with lorlatinib in a clinical setting in lung cancer patients with ALK rearrangement.

The work presented for review provides important information on the possible reduction of side effects of lorlatinib through the use of silibinin. Both the presented results and their discussion do not arouse any major objections.

Reviewer 3 Report

This manuscript describes the potential use of silibinin to reduce lorlatinib-associated dyslipidaemia. The manuscript design is confusing and the final confusion is not clear. The effects of silibinin on lipid uptake and the CYP3A4 seems like two separate focus.

Comments:

1. Missing figure 2

2. Did authors validate their LC-MS data using lipid standards? Any LC-MS data shown? Are the LC-MS data deposit in public database?

3. What is the mechanism of lorlatinib to increase lipid level?

4. Figure 4 is missing.

5. The CYP3A4 docking and validation experiments seems out of place. Any explanation on doing it instead on studying the mechanism of lipid lowering effects

6. Figure 5 n=3 is needed

Round 2

Reviewer 1 Report

No further comments.

Author Response

We wish to thank the reviewer for the kind and warm acceptance of our revised manuscript. 

Reviewer 3 Report

Fig. 1 Please increase the font size for the venn diagram. Also add dotted line to all the volcano plots to show your thresholds for differential metabolites

Author Response

According to the reviewer's suggestions, we have now increased the font size for the venn diagram in Fig. 1. We have also added dotted lines to all the volcano plots (Fig. 1) to better show the thresholds for differential metabolites.